# The Clinical Significance and Role of CXCL1 Chemokine in Gastrointestinal Cancers

**DOI:** 10.3390/cells12101406

**Published:** 2023-05-17

**Authors:** Jan Korbecki, Mateusz Bosiacki, Katarzyna Barczak, Ryta Łagocka, Dariusz Chlubek, Irena Baranowska-Bosiacka

**Affiliations:** 1Department of Biochemistry and Medical Chemistry, Pomeranian Medical University in Szczecin, Powstańców Wlkp. 72, 70-111 Szczecin, Poland; jan.korbecki@onet.eu (J.K.); mateusz.bosiacki@pum.edu.pl (M.B.); dchlubek@pum.edu.pl (D.C.); 2Department of Anatomy and Histology, Collegium Medicum, University of Zielona Góra, Zyty 28 St., 65-046 Zielona Góra, Poland; 3Department of Functional Diagnostics and Physical Medicine, Faculty of Health Sciences, Pomeranian Medical University in Szczecin, Żołnierska 54 Str., 71-210 Szczecin, Poland; 4Department of Conservative Dentistry and Endodontics, Pomeranian Medical University, Powstańców Wlkp. 72, 70-111 Szczecin, Poland; katarzyna.barczak@pum.edu.pl (K.B.); ryta.lagocka@pum.edu.pl (R.Ł.)

**Keywords:** chemokine, cytokine, CXCL1, tumor, GRO-α, MGSA, hepatocellular carcinoma (HCC), pancreatic ductal adenocarcinoma (PDAC), colon cancer

## Abstract

One area of cancer research is the interaction between cancer cells and immune cells, in which chemokines play a vital role. Despite this, a comprehensive summary of the involvement of C-X-C motif ligand 1 (CXCL1) chemokine (also known as growth-regulated gene-α (GRO-α), melanoma growth-stimulatory activity (MGSA)) in cancer processes is lacking. To address this gap, this review provides a detailed analysis of CXCL1’s role in gastrointestinal cancers, including head and neck cancer, esophageal cancer, gastric cancer, liver cancer (hepatocellular carcinoma (HCC)), cholangiocarcinoma, pancreatic cancer (pancreatic ductal adenocarcinoma), and colorectal cancer (colon cancer and rectal cancer). This paper presents the impact of CXCL1 on various molecular cancer processes, such as cancer cell proliferation, migration, and invasion, lymph node metastasis, angiogenesis, recruitment to the tumor microenvironment, and its effect on immune system cells, such as tumor-associated neutrophils (TAN), regulatory T (T_reg_) cells, myeloid-derived suppressor cells (MDSCs), and macrophages. Furthermore, this review discusses the association of CXCL1 with clinical aspects of gastrointestinal cancers, including its correlation with tumor size, cancer grade, tumor–node–metastasis (TNM) stage, and patient prognosis. This paper concludes by exploring CXCL1’s potential as a therapeutic target in anticancer therapy.

## 1. Introduction

Gastrointestinal tumors are a diverse group of cancers that affect organs responsible for digestion. These tumors are categorized based on their specific organ location, including head and neck cancer, esophageal cancer, gastric cancer, liver cancer, cholangiocarcinoma, pancreatic cancer, colon cancer, and rectal cancer. In 2021, there were an estimated 5.95 million new cases of these tumors, representing 30.9% of all cancer diagnoses [1]. Additionally, there were 4.06 million deaths caused by gastrointestinal tumors, accounting for 40.9% of all cancer-related deaths [1]. These high mortality rates highlight the need for more effective treatment options, which has led to increased research into potential therapeutic targets.

One promising area of research focuses on intercellular signaling within tumor tissue, specifically the interaction between cancer cells and the immune system [2,3,4,5,6,7]. Cytokines, extracellular signaling molecules that regulate various immune cells, play a critical role in this interaction [8]. Among cytokines, chemokines have chemotactic properties and are divided into four subfamilies based on a conservative motif at the N-terminus [9]. The CXC chemokine subfamily includes 16 representatives in humans, which are divided based on their ability to activate the CXCR receptors [9]. C-X-C motif receptor 2 (CXCR2) ligands, including C-X-C motif ligand 8 (CXCL8, interleukin-8 (IL-8)), are the most frequently studied chemokines, followed by CXCL1.

CXCL1 is a chemokine consisting of 73 amino acids and has a molecular weight of 8 kDa [10]. Its expression is regulated at both the transcription and CXCL1 mRNA stability levels [11,12]. This chemokine activates the CXCR2 receptor at concentrations of several nM [13], making CXCR2 its most significant receptor. At approximately 100-fold higher concentrations, CXCL1 can also activate the CXCR1 receptor [13]. However, the role of CXCR1 in the physiological and pathological functions of CXCL1 appears to be less significant.

Another receptor for CXCL1 is ACKR1 [14], though the importance of this receptor remains unclear. ACKR1 seems to regulate the availability of various chemokines, including CXCL1 [15], and may participate in the transport and distribution of CXCL1 within the intercellular space [16].

Activation of the CXCR2 receptor by CXCL1 triggers signal transduction. Heterotrimeric G proteins, particularly the inhibitory guanine nucleotide regulatory protein (Gα_i_), are directly activated by CXCR2 [17]. Intracellularly, many proteins bind directly to CXCR2 [18], playing a crucial role in signal transduction, with some signaling pathways operating independently of G proteins.

Activation of CXCR2 by CXCL1 induces cell migration. Among blood cells, neutrophils exhibit the highest expression of CXCR2, making CXCL1 an essential chemoattractant for neutrophils [9,19]. Furthermore, CXCL1 displays mitogenic properties, demonstrated on melanoma cells as one of the chemokine’s first identified properties. As a result, CXCL1 was initially referred to as melanoma growth-stimulatory activity (MGSA) [20].

Considerably less attention has been paid to other CXCR2 ligands in cancer research. As a result, knowledge of the significance of other CXCR2 ligands in cancer processes is incomplete. Moreover, there is a lack of available review articles that describe the significance of certain CXCR2 ligands in cancer processes, including CXCL1. With over 1300 papers available in the PubMed database on this chemokine in the context of cancer (https://pubmed.ncbi.nlm.nih.gov, accessed on 23 March 2023), it is difficult to obtain a clear picture of the current knowledge in this area. To address this gap, we have reviewed the significance of CXCL1 in tumors, with a focus on gastrointestinal tumors due to the sheer volume of available information.

## 2. Head and Neck Cancer

Head and neck cancer is a group of cancers located most often in the oral cavity, oropharynx, hypopharynx, nasopharynx, nasal cavity, and larynx [21,22]. Annually, approximately 890,000 new cases of head and neck cancer are diagnosed, with approximately 450,000 deaths [22]. The main form of this cancer is squamous cell carcinoma. Risk factors include cigarette smoking, heavy drinking, and oral infection by carcinogenic human papillomavirus (HPV) serotypes [23].

Head and neck squamous cell carcinoma [24,25,26,27], including oral squamous cell carcinoma (OSCC) [28,29,30,31] and larynx squamous cell carcinoma (LSCC) [32,33], show elevated CXCL1 expressions relative to healthy tissue. According to bioinformatics analysis, *CXCL1* is considered one of the hub genes with one of the highest number of protein-protein associations in oral cancer [34], particularly in OSCC [35] and oral tongue squamous cell carcinoma (OTSCC) [36]. In patients with head and neck cancer, blood CXCL1 levels may be lower than in healthy individuals [37].

CXCL1 is secreted by cancer cells in head and neck cancer [38,39] including OSCC cells [28,39]. This occurs as a result of interleukin-1β (IL-1β) [40] and vascular endothelial growth factor (VEGF) and vascular endothelial growth factor receptor 1 (VEGFR1) activation on cancer cells [41]. VEGFR1 causes an increase in B-cell leukemia/lymphoma-2 (Bcl-2) expression in cancer cells; this protein activates nuclear factor κB (NF-κB), the transcription factor responsible for the increase in CXCL1 expression, as well as CXCL8 [42]. Other secretory factors that can increase CXCL1 expression in head and neck cancer cells are interleukin-1α (IL-1α) and epidermal growth factor (EGF), as suggested by experiments in mouse models [43]. CXCL1 expression is also increased by elevated Snail2 expression in cancer cells [44]. In addition to cancer cells, cancer-associated fibroblast (CAF) also produces CXCL1 in OSCC [31]. CXCL1 expression in these cells is increased by IL-1α [45] and IL-1β [31]. OSCC cancer cells induce the senescence of fibroblasts [46]; in this state, fibroblasts secrete CXCL1, which, in an autocrine manner, enhances the senescence of fibroblasts and participates in tumorigenic processes in OSCC.

CXCL1 may contribute to the formation of head and neck cancer. The expression of this chemokine is elevated in fibroblasts in oral submucous fibrosis [25], the precancerous lesions from which head and neck cancer arises. CXCL1 promotes tumorigenic processes in oral submucous fibrosis; it causes the proliferation and migration of keratinocytes, as well as increasing the stemness of these cells [25]. These processes lead to the transition of oral submucous fibrosis into head and neck cancer.

CXCL1 is also involved in tumorigenesis in head and neck cancer. CXCL1 increases the proliferation of OSCC cancer cells [40,45,47], which is associated with the activation of CXCR2 that transactivates epidermal growth factor receptor (EGFR) [40].

CXCL1 also causes OSCC cancer cells to migrate [31,46,47]. Specifically, OTSCC cancer cells cause an increase in CXCL1 expression in lymphatic endothelial cells (LEC) [48], which leads to head and neck cancer cells migrating into LEC, and consequently cancer cells migrating into lymphatic vessels and lymph node metastasis. This may explain the correlation of high CXCL1 expression in the tumor with the presence of lymph node metastasis in head and neck cancer [28,32].

CXCL1 increases the expression and secretion of matrix metalloproteinase (MMP)7 and MMP9 by OSCC cells [47], which promotes tumorigenesis.

CXCL1 is involved in angiogenesis in OSCC tumors [28,41]. Following an increase in the expression of CXCL1 in cancer cells [41] and endothelial cells [49] as a result of VEGF, CXCL1 acts on endothelial cells, causing angiogenesis, and on cancer cells, causing their migration toward endothelial cells. This means that CXCL1 is involved in the reciprocal interaction of OSCC cancer cells with endothelial cells. CXCL1 may also be responsible for the recruitment of TAN into the head and neck cancer tumor niche [50].

The high expression of CXCL1 in OSCC [28] and LSCC [32,33] tumors is correlated with lymph node metastasis. Additionally, in LSCC, CXCL1 is associated with the tumor–node–metastasis (TNM) stage and is inversely correlated with tumor histopathological grade [33]. At the same time, CXCL1 expression is not associated with T classification, or LSCC tumor size [32]. In LSCC, CXCL1 is not associated with tumor differentiation [32]. In contrast, in OSCC tumors, CXCL1 is not associated with mode of invasion [28] or tumor differentiation [28].

High levels of CXCL1 in the blood of patients with head and neck cancer are associated with failure of radiation therapy [37]. Higher CXCL1 expression in the tumor is also associated with a worse prognosis of head and neck squamous cell carcinoma (Table 1) [26,27,31,51], including LSCC [33].

## 3. Esophageal Cancer

More than 604,000 new cases of esophageal cancer are diagnosed annually, which accounts for 3.1% of all diagnosed cancer cases [1]. Each year, there are approximately 544,000 deaths resulting from this cancer, which constitutes 5.5% of all cancer-related deaths [1]. Esophageal cancer can be divided into two subtypes: esophageal squamous cell carcinoma (ESCC) and esophageal adenocarcinoma [53]. ESCC accounts for 70% of all esophageal cancers. Its risk factors include cigarette smoking and high alcohol consumption, while for esophageal adenocarcinoma it is obesity and gastro-oesophageal reflux disease [53]. Frequent damage to the esophagus from gastric contents may lead to Barrett’s esophagus and then to esophageal adenocarcinoma [53].

CXCL1 expression is elevated in ESCC tumors relative to healthy tissue [54,55], along with an elevated level of CXCL1 in the blood compared to healthy individuals [56]. Bioinformatic analyses have identified *CXCL1* as one of the hub genes in ESCC [55,57]. A high level of CXCL1 in the blood is associated with a low risk of ESCC [58], which may reflect either antitumor mechanisms involving CXCL1 in the early stages of ESCC development or a protective effect. An elevated expression of CXCL1 has been observed in Barrett’s esophagus and esophageal adenocarcinoma compared to healthy tissue [59].

In ESCC tumors, CXCL1 is secreted by ESCC cells as a result of high NF-κB activation, mediated by several factors, for example, laminin subunit gamma 1 (LAMC1) [60] and early growth response-1 (EGR-1) [61]. By activating the receptor CXCR2, CXCL1 increases the expression of EGR-1, meaning that there is a positive feedback loop between EGR-1 and CXCL1 [61]. CXCL1 expression in ESCC cells is also increased by CAF [56]. It seems that CXCL1 in ESCC tumors comes mainly from CAF [56]. The expression of this chemokine in CAF is also increased by cancer cells.

Tumor-associated macrophages (TAM) are another source of CXCL1 in ESCC tumors [62]. CXCL1 expression is increased upon exposure of macrophages to secretory factors from ESCC cells, which results in the differentiation of these macrophages into TAM [62].

The increase in CXCL1 expression in Barrett’s esophagus and esophageal adenocarcinoma is also due to the frequent amplification of the *CXCL1* gene as well as hypomethylation of its promoter [59]. Barrett’s esophagus is accompanied by a decreased expression of glutathione peroxidase 7 (GPX7) [63]. Due to the fact that GPX7 inhibits tumor necrosis factor-receptor 1 (TNFR1) activation when GPX7 expression is downregulated, there is increased tumor necrosis factor-α (TNF-α) activity, resulting in inflammatory responses in the esophagus when exposed to gastric contents. This activates NF-κB and results in the increased expression of proteins dependent on this transcription factor, such as CXCL1. Chronic inflammation leads to esophageal adenocarcinoma.

CXCL1 is involved in tumorigenic processes in ESCC. CXCL1 increases the proliferation of ESCC cells [54]. Through the CXCR2, CXCL1 increases the expression of EGR-1 in ESCC cells [61], a transcription factor that increases their proliferation. This effect of CXCL1 on ESCC cell proliferation is autocrine [54], as ESCC cells secrete CXCL1 that then acts on the same cells.

CXCL1 is also responsible for radio-resistance in ESCC cells [56]. This property of CXCL1 is dependent on extracellular signal-regulated kinase (ERK) mitogen-activated protein kinase (MAPK) activation and decreased superoxide dismutase 1 (SOD-1) expression, which leads to an increase in the level of reactive oxygen species (ROS) in ESCC cells which increases the expression of DNA repair enzymes when cells are exposed to radiation therapy [56].

CXCL1 acts on CAF. CXCL1 causes a change in the phenotype of CAF into inflammatory CAF through activation of the CXCR2-signal transducer and activator of transcription 3 (STAT3) pathway [60]. These altered CAF secrete factors such as IL-1β, interleukin-6 (IL-6), leukemia inhibitory factor (LIF), and granulocyte colony-stimulating factor (G-CSF), which contribute to tumorigenesis. CXCL1 also causes the recruitment of granulocytic-myeloid-derived suppressor cells (G-MDSCs) to the tumor niche in ESCC tumors [64].

CXCL1 expression in ESCC tumors is correlated with tumor size. A larger tumor shows higher CXCL1 expression [56]. Immunohistochemical studies have shown that ESCC tumors larger than 6.5 cm show higher CXCL1 expression than in the sample of ESCC tumors smaller than 6.5 cm. CXCL1 expression is also higher in ESCC stages II–IV than in stage I, showing that the expression of CXCL1 increases with tumor growth and development [57]. At the same time, CXCL1 expression in ESCC is not associated with the depth of invasion or lymph node metastasis [56]. Higher CXCL1 expression in CAF in ESCC tumors may be associated with a poorer prognosis (Table 2) [56], and higher CXCL1 expression in tumors may be associated with a better prognosis for ESCC patients [57]. However, there are also studies that show no association between CXCL1 expression in the tumor and prognosis for esophageal cancer patients [65].

## 4. Gastric Cancer

It is estimated that nearly 1.06 million new cases of gastric cancer were diagnosed in 2020, which accounted for 5.6% of all cancer diagnoses [1]. Further, in 2020, nearly 770,000 gastric cancer deaths were diagnosed, which accounted for 7.7% of cancer deaths [1]. The main cause of gastric cancer is *Helicobacter pylori* infection [66,67]. It is estimated that nearly 4.4 billion of the world’s population were infected with this bacterium in 2015 [68]. The largest percentage of the population infected with this bacterium is in Africa and the smallest in Oceania. *H. pylori* causes chronic inflammation, which leads to gastritis, which after some time develops into intestinal metaplasia, atypical hyperplasia, and, finally, gastric cancer [69]. An essential virulence factor in causing gastric damage by *H. pylori* is cytotoxin-associated gene A (CagA) [70], a protein that is guided to gastric epithelial cells via the type IV secretion system (TIVSS) [71]. CagA causes chronic inflammation, which is the basis for the development of gastric cancer. However, it is not the only virulence factor in *H. pylori*. Other virulence factors relevant to the pathogenesis of gastric cancer caused by *H. pylori* also include outer membrane inflammatory protein A (OipA), duodenal ulcer promoting gene A protein (DupA), and vacuolating cytotoxin A (VacA) [72].

*H. pylori* infection and the effects of this bacterium cause chronic inflammation in the gastric wall, which leads to increased levels of pro-inflammatory cytokines such as TNF-α, C-C motif ligand 5 (CCL5, regulated on activation, normally T cell expressed and secreted (RANTES)), C-C motif ligand 3 (CCL3, macrophage inflammatory protein 1α (MIP-1α)), C-X-C motif ligand 9 (CXCL9, monokine induced by interferon-γ (MIG)), C-C motif ligand 20 (CCL20, liver and activation-regulated chemokine (LARC)) and CXCR2 ligands such as CXCL8, C-X-C motif ligand 2 (CXCL2, growth-regulated gene-β (GRO-β)) and the described CXCL1 [69,73,74]. *H. pylori* increases the expression of CXCL1, as well as other CXCR2 ligands, in gastric epithelial cells [71,73,74,75]. The increased expression of CXCR2 ligands is higher in gastric epithelial cells than in gastric cancer cells [76], which shows that CXCL1 may be significant in the early stages of tumor formation. At the same time, the expression level of CXCL1 is correlated with the concentration of *H. pylori* [73] and decreases following antimicrobial therapy [77].

The mechanism for the increase in CXCL1 expression involves multiple pathways. *H. pylori* causes the translocation of CagA and fragments of peptidoglycan via TIVSS into gastric epithelial cells [71], which provokes an increase in CXCL1 expression [76]. The bacterium can also be recognized by toll-like receptor 2 (TLR2) independently of the Cag pathogenicity island, which increases the expression of CXCL1 in gastric epithelial cells [75]. EGFR, ERK MAPK, c-Jun N-terminal kinase (JNK) MAPK, and janus tyrosine kinase (JAK)/STAT are also important in this mechanism [75,78]. At the same time, TLR2 activation reduces CXCL1 expression in conventional dendritic cells (cDCs) [79]. Another mechanism for increasing CXCL1 expression is TNF-α inducing protein (Tip-α), secreted by *H. pylori*. This factor causes increases in the expression of chemokines such as C-C motif ligand 2 (CCL2, monocyte chemoattractant protein 1 (MCP-1)), C-C motif ligand 7 (CCL7, monocyte chemoattractant protein 3 (MCP-3)), CCL20, C-X-C motif ligand 10 (CXCL10, γ interferon inducible protein 10 (IP-10)), and CXCR2 ligands [80], which are associated with the activation of NF-κB by Tip-α. *H. pylori* may also indirectly increase the expression of CXCL1 and CXCL8. *H. pylori* activates NF-κB in a CagA-dependent manner which results in the increased expression of interleukin-32 (IL-32) [81] which increases CXCL1 expression. An increase in CXCL1 expression, similar to other CXCR2 ligands, causes neutrophils to infiltrate the gastric foveolar epithelium [82,83]. These cells also secrete CXCL1 and CXCL8 [84]. This means that neutrophils cause the infiltration of more neutrophils in *H. pylori* gastritis. Neutrophils are involved in inflammatory responses in gastritis; they secrete ROS and reactive nitrogen species (RNS), which damage tissue and cause gene instability [83].

Chronic inflammation leads to gastritis, which after some time develops into gastric intestinal metaplasia, then atypical hyperplasia, and, finally, gastric cancer [69]. The progression of *H. pylori* infection to gastric cancer is associated with the CXCL1/CXCL8-CXCR2 and p53 axis. *H. pylori* infection increases the expression of CXCR2, the receptor for CXCL1 [74], which is associated with the activation of nuclear factor-κB subunit 1 (NF-κB1).

*H. pylori* infection also increases the expression of CXCL1 as well as other ligands for CXCR2 [69,73,74,76,82]. The increased activation of the CXCL1/CXCL8-CXCR2 axis causes cellular senescence of gastric cells, which leads to atrophic mucosa. Importantly, cellular senescence is a process that inhibits proliferation and thus inhibits gastric cancer development; this process creates a positive feedback loop between CXCR2 and p53 [74] where p53 increases CXCR2 expression, while CXCR2 increases p53 activation. With this mechanism, CXCR2 increases proliferation and p53 blocks proliferation. In the next stage of gastric cancer development, *H. pylori* interfere with p53 function and increase p53 expression in gastric mucosa [85,86,87,88]. A disorder in the function of p53 is associated with *H. pylori* causing DNA damage [89], which leads to mutations in the *TP53* gene [86,90]. This is partly due to the formation of the CagA complex with partitioning-defective 1b (PAR1b)/microtubule affinity-regulating kinase 2 (MARK2), which leads to a decrease in breast cancer type 1 susceptibility protein (BRCA1) function [91], which promotes DNA breakage and reduces DNA repair capabilities. *H. pylori* also disrupts the proper function of p53 by reducing the expression and function of upstream stimulatory factor 1 (USF1) and USF2, proteins that form a complex with p53 and are important in p53 function [92,93]. *H. pylori* decreases p53 protein expression by increasing degradation of the protein [87,94,95]—a result of the activation of human double minute 2 (HDM2)/mouse double minute 2 (MDM2) by CagA. The appearance of mutations in the *TP53* gene leads to escape from senescence [74] and p53 no longer inhibiting proliferation. Activation of CXCR2 by CXCL1 and CXCL8 stimulates the proliferation of transformed cells, which leads to gastric cancer.

Gastric cancer tumors show a higher expression of CXCL1 and CXCL8 than healthy gastric tissue [96,97,98,99]; the expression of both of these chemokines is highest in diffuse-type gastric carcinoma [100]. CXCL1 levels are positively correlated with patient age [101]. As one of the risk factors for developing gastric cancer is age [66], there may be a link between CXCL1, age, and gastric cancer. In addition, as serum levels of CXCL1 are elevated in gastric cancer patients [102,103], plasma CXCL1 levels may be a marker of gastric cancer.

The elevated expression of CXCL1 in gastric cancer tumors is associated with an increase in the expression of this chemokine in an autocrine manner in gastric cancer cells [104,105]. Another important cause of increased CXCL1 levels is *H. pylori* infection [82]. There is also downregulation of microRNA (miR)-204 in gastric cancer tumors [106]; a microRNA that directly downregulates CXCL1 expression. Another important, if not the most important, source of CXCL1 in gastric cancer tumors is TAM [107]. CXCL1 in gastric cancer tumors also comes from fibroblasts [108] which increase CXCL1 expression under the influence of extracellular vesicles from gastric cancer cells.

The positive correlation of CXCL1 expression with worse overall survival of gastric cancer patients is related to the fact that CXCL1 participates in tumorigenic processes. CXCL1 increases gastric cancer cell proliferation, as shown by experiments on HGC27 cells [99], AGS, Hs746T [104]. Further, CXCL1 causes cancer cell migration [98,99,105,107,109]. However, the CXCL1/CXCL8-CXCR2 axis may also cooperate with other chemokines in gastric cancer cell migration, particularly C-X-C motif ligand 12 (CXCL12, stromal-derived factor-1 (SDF-1))-CXCR4. Both receptors, CXCR2 and CXCR4, activate NF-κB and STAT3 [107,110]. NF-κB activation increases CXCR4 expression, while STAT3 activation increases CXCR2 expression. Therefore, activation of either of these two receptors results in an increased expression of CXCR2 and CXCR4, which translates into an increase in gastric cancer cell migration and epithelial-to-mesenchymal transition (EMT) induction of these cells [110], an effect also significant in metastasis. The CXCR2-STAT3 pathway may be involved in angiogenesis in gastric cancer [98]; activated STAT3 increases the expression of VEGF [98]—the most important growth factor that causes angiogenesis.

Gastric cancer cells secrete factors that increase CXCL1 expression in LEC [109,111] as a result of NF-κB activation in LEC [111]. CXCL1 activates the focal adhesion kinase (FAK)-ERK MAPK-RhoA pathway in LEC, which leads to the reorganization of F-actin and to the migration and tube formation of LEC. That indicates that CXCL1 in gastric cancer causes lymphangiogenesis.

CXCL1 induces an increase in the expression of MMP2 and MMP9 in gastric cancer [97,109] and causes the migration of LEC, especially to lymphatic vessels [109], which results in lymphatic metastasis. For this reason, CXCL1 levels in gastric cancer tumors are positively correlated with lymphatic metastasis [97,101,109,112]. Serum CXCL1 levels are also positively correlated with lymph node metastasis [102].

CXCL1 acts on tumor-associated cells. CXCL1 causes the recruitment of G-MDSCs [113], bone marrow-derived mesenchymal cells [114], and neutrophils [82] to the tumor niche (Figure 1). Bone marrow-derived mesenchymal cells in the tumor niche are transformed into myofibroblasts. On the other hand, CXCL1 levels are not correlated with the number of TAN in a gastric cancer tumor [100]. For this reason, it is possible that another CXCR2 ligand is responsible for recruiting neutrophils into the tumor niche.

Gastric cancer cells can increase CXCL1 expression in tumor-associated cells. Gastric cancer cells secrete TNF-α, which increases CXCL1 expression in TAM [107]. Further, gastric cancer cells secrete extracellular vesicles that alter the expression of various genes in different cells in the tumor niche, for example extracellular vesicles that contain miR-155, miR-193b and miR-210, inducing the expression of chemokines CXCL1 and CXCL8 in fibroblasts [108]. At the same time, this mechanism does not take place in myofibroblasts.

CXCL1 is closely associated with tumorigenesis in gastric cancer, as CXCL1 expression is positively correlated with advanced TNM stage, i.e., T invasion stage, lymph node metastasis, as well as tumor size [97,98,101,107,109,112,115]. Likewise, serum CXCL1 levels are positively correlated with tumor stage and lymph node metastasis [97,102].

Higher CXCL1 expression in gastric cancer tumors is associated with worse overall survival (Table 3) [97,98,99,101,105,109,112,115], although there is also one study where high CXCL1 expression was associated with a better prognosis for gastric cancer patients [96]. In plasma, higher CXCL1 levels are also associated with poorer cumulative survival [97]. After anticancer treatment, high blood levels of CXCL1 are an indicator of future recurrence [116].

## 5. Liver Cancer

The most common liver cancer is HCC [117]. An estimated 782,000 new cases of HCC are diagnosed annually, meaning the incidence of this cancer is at 10 cases per 100,000 population per year. There are also 746,000 deaths caused by this cancer each year. Risk factors for HCC include [117]:Chronic hepatitis B virus (HBV) infection;Chronic hepatitis C virus (HCV) infection;Consuming large amounts of alcohol, which leads to alcoholic liver disease (ALD), then to liver cirrhosis, and eventually HCC;Obesity, which leads to non-alcoholic fatty liver disease (NAFLD), then liver cirrhosis, and, eventually, HCC.

Due to the significance of chronic inflammation in the liver, CXCL1 plays an important role in the onset and development of HCC.

CXCL1 is important in the emergence of liver cancer, as demonstrated by studies on genetic polymorphisms in patients with liver cirrhosis and HCC. Specifically, more patients with liver cirrhosis and HCC have the *CXCL1* rs4074 A allele [118,119], implying that this allele is associated with a predisposition to these diseases. Individuals with this variant have higher levels of CXCL1 in their blood than those with the rs4074 G variant [119]. Further, the production of CXCL1 in response to HCV proteins is higher in individuals with this variant of the *CXCL1* gene [118].

During exposure of the liver to agents leading to liver cirrhosis and HCC, NF-κB is activated and CXCL1 expression increases. In liver cells, there is p50:p50 NF-κB [120], which does not cause an increase in CXCL1 expression, but blocks changes in the expression of this chemokine. This is due to the simultaneous recruitment of p50:p50 NF-κB and histone deacetylase 1 (HDAC1) to the CXCL1 promoter [120]. This mechanism inhibits the increased expression of CXCL1 and also CXCL2, S100 calcium binding protein A8 (S100A8) and S100 calcium binding protein A9 (S100A9), which suppresses the formation of liver steatosis and HCC. Nevertheless, with chronic liver inflammation, this mechanism only delays the development of the disease.

Liver steatosis and HCC arise from chronic inflammation of the liver. This condition can be caused by either chronic HBV or HCV infection. HBV causes an increase in CXCL1 production in the liver. The hepatitis B X (HBx) antigen binds to the *TGFB1* gene, which increases the expression of TGF-β1 in hepatocytes [121], a cytokine that activates hepatic stellate cells and more specifically increases the expression of CD147 in these cells [122]. This results in an increase in CXCL1 expression in hepatic stellate cells [123].

Further, chronic HCV infection is associated with increased CXCL1 expression in the liver. During HCV infection, hepatocytes produce CXCL1 [118]. Hepatic stellate cells increase CXCL1 production in HCV-infected hepatocytes [118].

Another cause of liver cirrhosis and HCC is frequent alcohol consumption. Alcohol causes an increase in CXCL1 expression in hepatocytes [124]. Further, CXCL1 expression in the liver of ALD patients is higher than in healthy individuals [125,126]. CXCL1 acts on hepatic stellate cells [123], which causes an increase in α smooth muscle actin (αSMA) and α1(I) collagen expression in these cells, leading to liver fibrosis [123,127].

CXCL1 can also cause liver fibrosis through another pathway. CXCL1 causes infiltration of the liver by neutrophils, which activate hepatic stellate cells via ROS. Then hepatic stellate cells start producing collagen, which leads to hepatic fibrosis [128], which is significant during alcoholic hepatitis as well as other liver diseases. For this reason, CXCL1 expression in the liver is positively correlated with liver fibrosis, for example, in patients with HCV infection [129]. On the other hand, another study showed that CXCL1 levels in the blood are negatively correlated with ISHAK fibrosis score [130], while CXCL8 levels in blood were positively correlated with this parameter.

Liver fibrosis is associated with liver cirrhosis [117]. Liver cirrhosis is a risk factor for HCC. This cancer, similar to liver cirrhosis, is characterized by inflammatory responses in which CXCL1 plays a significant role.

CXCL1 expression is higher in HCC tumors than in healthy liver tissue [131], and is higher in HCC than in liver cirrhosis [132]. In patients with liver cancer, blood CXCL1 levels are elevated compared to healthy individuals [132] and those with liver cirrhosis [132]. Blood CXCL1 level is also elevated in hepatitis B-related HCC, and for this reason may be used as a biomarker of this cancer [133].

A significant source of CXCL1 in HCC is myofibroblast-like cells [134], which also produce other secretory factors such as CXCL8 and IL-6 [134]. CXCL1 expression occurs in cancer cells [135,136], and CXCL1 and CXCL8 expression occurs in HCC cancer stem cells [137]. The expression of both chemokines is upregulated by neurotensin (NTS) through ERK MAPK activation, with it being CXCL8 that acts in an autocrine manner on cancer stem cells [137], increasing their proliferation. CXCL1 does not act on HCC cancer stem cells, only on other cells in the tumor niche.

In HCC tumors, many factors increase CXCL1 expression in cancer cells. In particular, CXCL1 itself is responsible for increases in NF-κB activation and thus its own expression in an autocrine manner [138]. Other factors that increase CXCL1 expression include chronic hypoxia [135], EGFR activation [139], interleukin-17 (IL-17) [140], saturated fatty acid (SFA) including palmitate [141], tumorigenesis-related downregulation of both solute carrier family 7 member 2 (SLC7A2) [142], and receptor-interacting protein kinase-3 (RIPK3) [136]. CXCL1 expression in HCC also depends on microRNA. It has been shown that miR-200a [143], a microRNA that directly downregulates CXCL1 expression, is downregulated in this tumor [143], and thus increases CXCL1 expression in HCC tumors.

Studies in mice have indicated that the elevated expression of CXCR2 ligands in the liver facilitates HCC formation in this organ [144]. In mice, the expression of CXCR2 ligands in the liver is highest a few days after birth. This indicates that CXCR2 ligands may create a favorable microenvironment for the development of pediatric liver cancer, particularly hepatoblastoma [144]. However, these results need to be confirmed on a human model.

CXCL1 increases the proliferation of HCC cells [131,138,145]. Studies on mice indicate that CXCL1 can affect HCC cancer stem cells, as CXCR2 ligands cause quiescence of these cells. These chemokines induce mechanistic target of rapamycin complex 1 (mTORC1) activation, which inhibits the differentiation of HCC cancer stem cells [146].

Further, mouse CXCR2 ligands increase the number of HCC cancer stem cells—cells that are resistant to chemotherapeutics, such as doxorubicin [146]. Therefore, CXCR2 ligands increase the resistance of HCC to treatment; nevertheless, studies need to be repeated in human models to see whether a human CXCR2 ligand is responsible for this process.

CXCL1 is also important in HCC tumor growth [138]. CXCL1 induces migration and EMT of HCC cells [131,143]. This property of CXCL1 is dependent on NF-κB activation [131]. CXCL1 also induces angiogenesis in this cancer [140].

CXCL1 also acts on tumor-associated cells. This chemokine induces the recruitment of neutrophils [147] and G-MDSCs [148] into the HCC tumor niche. This action of CXCL1 should be understood as part of a broader meshwork of tumor-associated mechanisms. CXCL1 induces the recruitment of neutrophils into the HCC tumor niche [147]; these cells secrete CCL2 and C-C motif ligand 17 (CCL17, thymus and activation-regulated chemokine (TARC)), which induce the recruitment of monocytes and T_reg_, respectively [149]. Monocytes differentiate into TAM, which secrete pro-angiogenic VEGF [150], while T_reg_ participate in cancer immunoevasion [6].

The great importance of CXCL1 in tumorigenesis is confirmed by the correlation of CXCL1 expression with various clinical parameters of the cancer in question. The level of CXCL1 expression increases with HCC tumor growth. There is a positive correlation of CXCL1 expression in the tumor with TNM classification stages, macrovascular invasion, microvascular invasion, and distant metastasis [131,143]. Higher CXCL1 expression in the tumor is associated with a worse prognosis for liver cancer patients [151], including HCC (Table 4) [131,143,147,152].

## 6. Cholangiocarcinoma

Cholangiocarcinoma is a tumor located in the biliary tree [153]. The incidence of this type of cancer varies by world region. For the US, it is at 0.72 to 1.67 per 100,000 population. In contrast, the highest incidence is in Northeast Thailand, with 80 cases per 100,000 population [153]. Cholangiocarcinoma can be divided by tumor location into intrahepatic cholangiocarcinoma and extrahepatic cholangiocarcinoma. Factors that increase the likelihood of cholangiocarcinoma include sclerosing cholangitis, liver cirrhosis, *Opisthorchis viverrini* infection, HBV or HCV infection, and heavy alcohol consumption [153,154]. Studies of patients with cholangiocarcinoma have shown that this is one of the few cancers in which CXCL1 may have anticancer properties.

CXCL1 inhibits the proliferation of OCUG-1 and HuCCT1 cholangiocarcinoma cells [155] and does not affect the proliferation of KMBC cells [156]. CXCL1 also inhibits the migration of cholangiocarcinoma cancer cells [155]. The source of CXCL1 in cholangiocarcinoma tumors may be mesenchymal stem cells (MSC), as cancer cells secrete extracellular vesicles that act on MSC, causing their fibroblastic differentiation and increasing the expression of secretory factors such as CXCL1, CCL2, and IL-6 [156].

CXCL1 may also have pro-cancer properties in cholangiocarcinoma. Liver CXCR2 ligands regulate cancer immune evasion in cholangiocarcinoma, which originates in the gut. Disruption in gut barrier function leads to the passage of commensal gut bacteria into the liver, which activates toll-like receptor 4 (TLR4) on cells in that organ. This leads to an increase in the expression of CXCR2 ligands in the liver, as shown by experiments in mice [157]. Chronic inflammation, in particular chronic upregulation of CXCR2 ligand expression in the liver, leads to the accumulation of MDSCs in this organ [157]. As cells have immunosuppressive properties, and therefore this leads to cancer immune evasion in cholangiocarcinoma and promotes the development of this cancer. Nevertheless, this has only been shown in mouse experiments and therefore we need to study this cancer mechanism on human models to show which human CXCR2 ligand is responsible for this process.

CXCL1 expression does not increase with tumor stage [155]. The level of CXCL1 expression in the tumor is negatively correlated with distant metastasis, and higher CXCL1 expression in the tumor is a better [155] or worse prognosis for patients [65]. Higher CXCR2 expression is also associated with a better prognosis for patients with cholangiocarcinoma (Table 5) [155]. On the other hand, one study indicates that increased CXCR2 expression in the tumor is associated with a worse prognosis for patients with intrahepatic cholangiocellular carcinoma [158].

## 7. Pancreatic Cancer

In 2020 alone, more than 495,000 new cases of pancreatic cancer were diagnosed, accounting for 2.6% of all cancers [1]. This cancer has a high mortality rate, with only 4% of patients surviving 5 years after diagnosis [159]. In 2020, there were 466,000 such deaths, which accounted for 4.7% of deaths caused by all cancers [1]. In pancreatic cancer, mutations occur during tumorigenesis, in particular in the Kirsten rat sarcoma viral oncogene homologue (KRAS) oncogene, increasing the activity of this protein [159,160,161,162]. Mutations also occurred in the *CDKN2A*, *TP53*, *SMAD4*, and *BRCA2* genes [159,162]. This leads to the formation of a tumor.

In pancreatic cancer tumors [163], particularly in pancreatic ductal adenocarcinoma, there may be an upregulation of CXCL1 expression relative to healthy tissue [164,165,166]. However, other studies have shown that in humans, CXCL1 expression in pancreatic ductal adenocarcinoma tumors is not different relative to healthy tissue [167].

CXCL1 expression is found in pancreatic cancer cells [168,169]. Mutation in the *KRAS* gene, common in pancreatic cancer [160,161,162], leads to an increase in CXCL1 expression as well as other CXCR2 ligands [170,171]. Other factors cause an increase in CXCL1 expression in pancreatic tumor cells. In particular, an increase in CXCL1 expression may be induced by the elevated expression of p63 [169], a protein belonging to the p53 transcription factor family, with similar properties to p53 [172]. p63 binds to the promoter of the *CXCL1* gene which increases the expression of CXCL1. In a cancer cell, CXCL1 expression may also be due to the interaction of this cell with other non-cancerous cells in the tumor niche. CXCL1 expression in cancer cells can be increased by IL-17 [173] and also apolipoprotein E (ApoE) secreted by inflammatory CAF into the tumor microenvironment [174]. ApoE activates the low-density lipoprotein receptor (LDLR) on the tumor cell, which leads to an increase in NF-κB activation and an increase in CXCL1 expression in that cell [174].

CXCL1 expression in cancer cells may also depend on pancreatic stellate cells found in a healthy pancreas that are recruited to the cancer niche during pancreatic cancer tumor development. These cells are an essential component of the cancer stroma in pancreatic cancer and play an important role in its tumorigenesis [175]. Pancreatic stellate cells secrete exosomes that increase CXCL1 expression in pancreatic cancer cells [176]. Increased CXCL1 expression in pancreatic cancer tumors may also be caused by a disruption in the function of metaplastic tuft cells, which are solitary chemosensory cells [177].

Another source of CXCL1 in pancreatic tumors is inflammatory CAF [178,179]; in pancreatic cancer, these are stromal fibroblasts [167]. CAF in pancreatic tumors originate mainly from resident pancreatic stellate cells [180]. CAF can be divided into two types based on their phenotype: inflammatory CAF and myofibroblastic CAF. If CAF come into direct contact with cancer cells, they are transformed into myofibroblastic CAF [179]. Polarization of CAF to inflammatory CAF occurs in cells that do not come into direct contact with pancreatic cancer cells [179]. Under the influence of secretory factors from cancer cells, the expression of sequestosome-1 (sqstm1) in CAF is reduced [180], which leads to increased levels of ROS in CAF cells and results in their senescence and the acquisition of a senescence-associated secretory phenotype (SASP) by these cells [180,181,182]. The factor responsible for this process may be IL-1α secreted by cancer cells [169]. IL-1α expression in the cancer cell is dependent on p63. IL-1α enhances inflammatory CAF and downregulates the myofibroblastic CAF phenotype. Subsequently, inflammatory CAF secrete factors such as CXCL1 that help maintain their phenotype [183].

CXCL1 expression in the tumor may also be increased by recruited monocytes [184]. In particular, macrophages produce significant amounts of CXCL1 at the metastatic site of pancreatic ductal adenocarcinoma [185]. Monocytes produce CXCL1 and CXCL8 under the influence of IL-35, which is associated with activation by IL-35 of the GP130:IL12RB2 heterodimer, which activates the pSTAT1:pSTAT4 heterodimer. The pSTAT1:pSTAT4 complex attaches to the promoters of the aforementioned chemokines, thus increasing their expression.

CXCL1 is significant in the early stages of pancreatic cancer transformation. Very often mutations in the *KRAS* gene occur at the beginning of tumorigenesis [171], followed by oncogene-induced senescence [186] which involves the activation of NF-κB which increases the expression of CXCR2 ligands such as CXCL1 [171,186]. This means that such a cell is in a senescent state and exhibits the senescence-associated secretory phenotype (SASP) [181,182]. Increased secretion of CXCR2 ligands results in the activation of this receptor. At an early stage of tumorigenesis, this induces inhibition of the proliferation of such an altered pre-cancerous cell and prevents tumor development [186]. Further, the activation of NF-κB leads to the expression of factors that have pro-inflammatory effects, including an increase in the number of antitumor macrophages with M1 polarity. In the early stages of tumorigenesis, p53 and p16/retinoblastoma (Rb) pathways are responsible for inhibiting proliferation and further development of pancreatic cancer. When mutations occur in the *TP53* gene, a senescence bypass occurs [186], an event common during tumorigenesis in pancreatic cancer as approximately 76% of pancreatic tumors show a mutation in *TP53* [162]. Then, activation of CXCR2 by CXCL1 causes the proliferation of the transformed cell and thus further development of pancreatic cancer [171].

In advanced pancreatic ductal adenocarcinoma tumors, CXCL1 also has pro-tumorigenic properties. Other CXCR2 ligands may also either be responsible or co-responsible for the properties of CXCL1. In particular, this refers to C-X-C motif ligand 5 (CXCL5, epithelial neutrophil-activating protein 78 (ENA78)) and CXCL8, which expression in pancreatic ductal adenocarcinoma tumors are the highest among CXCR2 ligands [166]. The low importance of CXCL1 in the development of pancreatic cancer tumors has been shown in animal experiments [167] and it has been suggested that this chemokine is more important in pancreatic cancer metastasis [167].

This effect of CXCL1 on pancreatic cancer cells can be negated if they have a high expression of atypical chemokine receptor 1 (ACKR1, Duffy antigen receptor for chemokines (DARC)) [187], an atypical receptor for various chemokines including CXCL1 [188] which inhibits the activity of CXCR2 and thus the action of CXCL1. Through activation of the receptor CXCR2, CXCL1 increases pancreatic ductal adenocarcinoma proliferation [171]. This effect may still be weak as in some studies CXCR2 did not affect the proliferation of pancreatic ductal adenocarcinoma cells [189] and was only weakly involved in the tumor growth of this cancer [167].

However, CXCL1 may be important in other cancer processes. CXCL1, together with other CXCR2 ligands, causes angiogenesis [170,173,190], and it may also be important in the metastasis of pancreatic cancer. In addition, this chemokine is important in cancer cell survival in lung metastasis [173]. One study shows that *CXCR2* gene knockout in mice leads to increased liver metastasis arising from pancreatic ductal adenocarcinoma [190]. As another available study shows opposite results [167], CXCL1 may have a role in metastasis only to certain organs and in certain models.

CXCL1 induces the recruitment of neutrophils into the tumor niche of pancreatic ductal adenocarcinoma [164,169,185] and G-MDSCs [177] but not monocytes and macrophages [167]. CXCL1 enhances the polarization of M2 macrophages [191,192,193,194] which inhibit the antitumor immune response more strongly than TAM not exposed to CXCL1. At the same time, neutrophils recruitment to pancreatic ductal adenocarcinoma is in balance with macrophage recruitment. When TAM recruitment is reduced by decreasing CCL2 expression in the tumor niche, there is an increase in CXCR2 ligand expression, which leads to increased neutrophils recruitment [164].

Further, CXCL1 can act on CAF where it enhances inflammatory CAF, decreases myofibroblastic-CAF phenotype [183], and lowers the expression of αSMA and production of ECM proteins. Inflammatory CAF also secrete increased amounts of various secretory factors, such as CXCR2 ligands [183]. This is associated with NF-κB activation in these cells. CXCL1 also causes an increase in connective tissue growth factor (CTGF) expression and secretion from CAF [195]. This growth factor is involved in tumorigenesis [196], causing proliferation and invasiveness of cancer cells, and participates in angiogenesis.

CXCL1 is also involved in chemotherapy resistance in pancreatic cancer patients. Some anticancer drugs, such as gemcitabine, increase the expression of CXCL1 and CXCL2 in pancreatic ductal adenocarcinoma tumor cells [185,192]. This has important implications for the recurrence of this cancer following chemotherapy with gemcitabine, where an increase is shown in the expression of CXCR2 ligands, including CXCL1, in pancreatic ductal adenocarcinoma [192] and in the metastatic site of this tumor [185]. This is associated with the increased expression of RIPK1 and RIPK3, the components of necroptosis [192]. CXCL1 increases M2 polarization of macrophages [191,192,193,194], which have immunosuppressive properties and inhibit the antitumor immune system response augmented by chemotherapy. Increased CXCL1 expression also recruits neutrophils to the metastasis site; these cells secrete growth arrest specific 6 (Gas6) [185], a secretory factor activating the Axl receptor tyrosine kinase that causes tumor cell proliferation and thus the regrowth of the metastatic site following chemotherapy.

CXCL1 expression in tumors increases with the tumor stage [165]. CXCL1 expression level in tumor cells is positively correlated with N stage, T stage, and distant metastasis, but not with perineural invasion [163]. CXCL1 expression in tumor stroma is positively correlated with perineural invasion, N stage, and T stage [163]. The high importance of CXCL1 in tumor processes is reflected in the impact of this chemokine on patient prognosis. Higher CXCL1 expression in a tumor means a worse prognosis for patients (Table 6) [163]. However, there are also studies that show no association between CXCL1 expression in the tumor and prognosis for pancreatic cancer patients [65,165].

## 8. Colorectal Cancer

Colorectal cancer is one of the three most commonly diagnosed cancers. An estimated 1.88 million new cases of this cancer were diagnosed in 2020 alone, which accounted for 9.8% of all cancers diagnosed [1]. Further, there were nearly 916,000 deaths caused by this cancer in 2020, which accounted for 9.2% of deaths caused by all cancers. Risk factors for this type of cancer include genetic factors and diet [197]. Genetic predisposition accounts for 10% to 20% of colorectal cancer cases [197]. Further, infection with certain intestinal bacteria may be associated with an increased likelihood of developing colorectal cancer [198]. As this cancer is associated with inflammation, one of the components of its pathogenesis is CXCL1.

There is elevated CXCL1 expression in colorectal cancer tumors compared to healthy tissue [199,200,201,202,203,204,205,206,207,208,209,210]. In addition, CXCL1 may also be a marker of early-stage colorectal cancer [211]. However, serum CXCL1 levels are reduced in patients with colorectal cancer [212]. In contrast, serum CXCL8 levels are increased in the same patients [212]. Some studies show that CXCL1 expression is higher in colon cancer tumors of elderly patients older than 65 years compared to patients younger than 65 years [205,213], and at least one study shows an inverse relationship [214].

In colorectal cancer, CXCL1 is produced by a variety of cells [215,216,217]. In the tumor niche, myofibroblasts [218], tumor-associated dendritic cells (TADC) [219], and TAM [220] may also be responsible for CXCL1 production, with various factors responsible for CXCL1 expression in these cells. In approximately 35% of colorectal cancer tumor cases, SMAD family member 4 (SMAD4) expression is reduced [212]. As SMAD4 reduces CXCL1 expression by decreasing inhibitor of NF-κB kinase β (IKKβ) activity and affecting glycogen synthase kinase-3β (GSK-3β), a decrease in SMAD4 expression in a tumor cell increases CXCL1 expression. An elevated CXCL1 expression in a colorectal cancer tumor may also be due to hypoxia-inducible factor-2 (HIF-2) which has been shown to be responsible for the expression of CXCR2 ligands in colorectal tumors in a murine model [221].

Other factors increasing CXCL1 expression include IL-22, secreted by CD4^+^ and CD8^+^ polyfunctional T cells [222], and cell migration inducing hyaluronidase 1 (CEMIP, KIAA1199), a protein that acts on hyaluronic acid and activates transforming growth factor-β receptor 1 and 2 (TGFBR1/2) which stimulates CXCL1 expression [223]. CXCL1 expression in colorectal cancer cells is also increased by prostaglandin E_2_ (PGE_2_) [215,216] in a process that depends on the EGFR-MAPK pathway. Further, mutation in Ras proteins can cause an increase in the activity of these proteins and lead to an increase in CXCL1 expression in colorectal cancer cells [217,218]. Elevated CXCL1 expression in colorectal cancer cells may also be the result of high basal NF-κB activation [224] associated with the overexpression of *UEV1A*, an enzyme that catalyzes poly-ubiquitination, which results in the activation of NF-κB. This transcription factor directly attaches to the CXCL1 promoter and then increases the expression of this chemokine.

Increased CXCL1 expression in colorectal cancer cells may also be associated with adiponectin [225,226]. CXCL12-CXCR4 axis [227], and microRNAs, e.g., miR-302e in colorectal cancer [208], miR-145-5p in colorectal cancer [228], and miR-1-3p in rectal adenocarcinoma [229], whose expression is decreased in colorectal cancer tumors. These microRNAs directly reduce CXCL1 expression. That is, the downregulation of the expression of these microRNAs is followed by an increase in CXCL1 expression.

CXCL1 in colorectal cancer tumors participates in tumorigenic processes. Due to links in tumor mechanisms with other proteins, *CXCL1* is considered a hub gene in colorectal cancer [229,230]. CXCL1 increases cancer cell proliferation [104,201,208,231,232] and increases glycolytic enzyme expressions, such as glucose transporter 1 (GLUT1), hexokinase 2 (HK2), and lactate dehydrogenase A (LDHA) [233]. In addition, in colorectal cancer cells, and in fibroblasts it decreases the expression of fibulin-1, a matrix protein that characterizes normal intestinal tissue [201]—the reduced expression allows colorectal cancer to develop.

CXCL1 also causes the migration and invasion of colorectal cancer cells [208,219,231,232], which is associated with an increase in MMP7 expression, and CXCL1 causing EMT of cancer cells [219]. For this reason, CXCL1 is associated with so-called “tumor budding” [234]—the appearance of either single cancer cells or small clusters of cancer cells at the invasive front of tumors [235].

CXCL1 may also facilitate the formation of metastasis by inducing an increase in parathyroid hormone-like hormone (PTHLH) expression in colorectal cancer cells [219]—a factor that causes bone remodeling during the formation of bone metastasis. CXCL1 is also carried with blood from colorectal cancer cells to the liver where it causes the infiltration of MDSCs to this organ by [220]. These cells form a pre-metastatic niche in the liver which facilitates the metastasis to this organ. High levels of CXCL1 in the blood are correlated with levels of circulating tumor cells [236], which indicates that CXCL1 is associated with a high likelihood of colorectal cancer metastasis.

CXCL1 also affects cells in the tumor niche in colorectal cancer. CXCL1 causes the recruitment of neutrophils [50,222,223] and G-MDSCs [216,237] into the tumor niche. The latter are cells with immunosuppressive properties that inhibit the antitumor immune response; in particular, they reduce the effect of cytotoxic CD8^+^ T cells. In addition, CXCL1 in colorectal cancer tumors causes angiogenesis [215] and stromal fibroblast senescence [226] which alters the secretory phenotype of these cells and thus enhances cancer tumor growth.

The level of colorectal tumor development cannot be definitely linked to the level of CXCL1 expression in the tumor, as some studies show a positive correlation and some a negative correlation between CXCL1 levels and colorectal tumor grade. Some studies show that CXCL1 expression in colorectal tumors is positively correlated with lymph node metastasis, tumor size, and tumor stage [233,238,239]. A higher level of CXCL1 in the blood of a colorectal cancer patient is associated with a higher TNM stage and may be an indicator of lung metastasis [236,240]. However, there are also studies where the elevated CXCL1 expression in a colorectal tumor is inversely correlated with lymph node metastasis [214,218] or tumor stage [202,207]. Some studies on colon cancer have shown that the higher the tumor stage is, the lower the level of CXCL1 in the tumor [205,209,213]. Finally, CXCL1 expression in liver metastasis is lower than in the primary colorectal tumor [202]. The cited results demonstrate that different research groups report negative and positive correlations of CXCL1 expression in the tumor with tumor stage, i.e., contradictory results. The results of studies on the relationship between CXCL1 levels in the tumor and prognosis for the colorectal cancer patient are varied. Some show that a higher CXCL1 expression in the tumor is associated with a worse prognosis for the patient [233,238]. Further, the prognosis is worse for rectal adenocarcinoma patients with a high CXCL1 expression in the tumor [229]. Other studies have shown that CXCL1 is associated with a worse prognosis only in patients with stage III colorectal cancer [204] or only in stage IV [218]. Still, other studies have found no association of CXCL1 expression in colorectal cancer tumors [206,239] and in blood [212] with patient prognosis (Table 7). Finally, there are also articles that indicate that high CXCL1 expression in the tumor is associated with a better prognosis for the colon cancer patient [209,213,241].

## 9. CXCL1 as a Therapeutic Target in Anticancer Therapy of Gastrointestinal Tumors

As described above, CXCL1 plays a significant role in the molecular processes of gastrointestinal tumors. In theory, it is possible to develop anticancer therapy targeting CXCL1. However, the most important receptor for CXCL1 is CXCR2 [9], which is also activated by other CXC chemokines, including CXCL2, CXCL3, CXCL5, CXCL6, CXCL7, and CXCL8. Therefore, a better therapeutic approach would be to target CXCR2 instead of CXCL1, which would block the effects of not only CXCL1 but also the other CXC chemokines listed above.

The most well-known and commonly tested CXCR2 antagonist as an antitumor agent is SB225002 (N-(2-hydroxy-4-nitrophenyl)-N’-(2-bromophenyl)urea) [242]. This compound has demonstrated antitumor activity, reducing the viability of SCC158 and HN30 oral squamous cell carcinoma cell lines [243]. It has also been shown to have no toxicity to the immortalized keratinocyte lineage HaCaT. In addition, SB225002 reduced the proliferation and migration of RBE and SSP25 intrahepatic cholangiocellular carcinoma cell lines in vitro [158]. Moreover, SB225002 inhibited the growth of cholangiocellular carcinoma tumors in mice [158]. However, it should be noted that SB225002 is not only a CXCR2 inhibitor but also binds to β-tubulin [244,245], leading to destabilization of microtubules and antimitotic activity of SB225002.

Additionally being tested are other CXCR2 antagonists that also inhibit CXCR1 activity. CXCL1 only activates CXCR2 at low concentrations, while CXCL8 activates both CXCR1 and CXCR2 at low concentrations [246]. Therefore, the use of dual CXCR1/CXCR2 antagonists allows for the inhibition not only of CXCL1 and other CXCR2 ligands but also of CXCL8. SCH-527123 is one such compound being tested [247,248]. SCH-527123 inhibits the proliferation and migration of colorectal cancer cell lines HCT116 and Caco2 in vitro [249]. SCH-527123 also sensitizes these cells to anticancer drugs, as demonstrated by experiments involving oxaliplatin [249]. In vivo studies in mice have confirmed that SCH-527123 exhibits anticancer activity and sensitizes colorectal cancer cells to oxaliplatin [249].

CXCR2 antagonists are currently under investigation in clinical trials as potential therapeutic agents (Table 8). On the ClinicalTrials.gov website (https://clinicaltrials.gov/ct2/home, accessed on 5 May 2023), a search using the receptor name “CXCR2” revealed 29 distinct clinical trials involving CXCR2 antagonists. Additional search results appear when entering the names of specific drugs.

Typically, a particular CXCR2 antagonist is examined for its efficacy in treating a specific group of diseases. For instance, SX-682 is being studied as a potential anticancer agent, while Ladarixin is being explored as a treatment for type 1 diabetes. Danirixin (GSK1325756), on the other hand, is being tested for lung diseases such as COPD and influenza. AZD5069 is under investigation as both an anticancer drug and a treatment for lung diseases such as COPD and asthma. Meanwhile, SB656933 is being examined as a potential therapeutic option for COPD and cystic fibrosis.

## 10. Future Perspectives

Among CXCR2 ligands, research predominantly focuses not on CXCL1 but on another CXCR2 ligand, CXCL8. As of 5 May 2023, the PubMed database (https://pubmed.ncbi.nlm.nih.gov) contained over 6-fold more scientific publications discussing CXCL8 in the context of cancer compared to CXCL1. This testifies to significant knowledge gaps remain regarding the role of CXCL1 in the oncogenic mechanisms of gastrointestinal neoplasms. Future research should also explore the interrelationships among individual CXCR2 ligands in tumorigenic processes. In humans, there are seven distinct CXCR2 ligands, which exhibit similar functions; however, differences among them warrant further investigation.

## 11. Conclusions

The role of CXCL1 in gastrointestinal malignancies has been extensively investigated and numerous studies have assessed the clinical implications of this chemokine in these tumors. Additionally, research has demonstrated the involvement of CXCL1 in a multitude of molecular mechanisms associated with these neoplasms. Presently, clinical trials exploring anticancer therapies incorporating CXCR2 antagonists are in preliminary phases. In the forthcoming years, these therapeutic agents are expected to become included in widely available cancer treatment.

## Figures and Tables

**Figure 1 cells-12-01406-f001:**
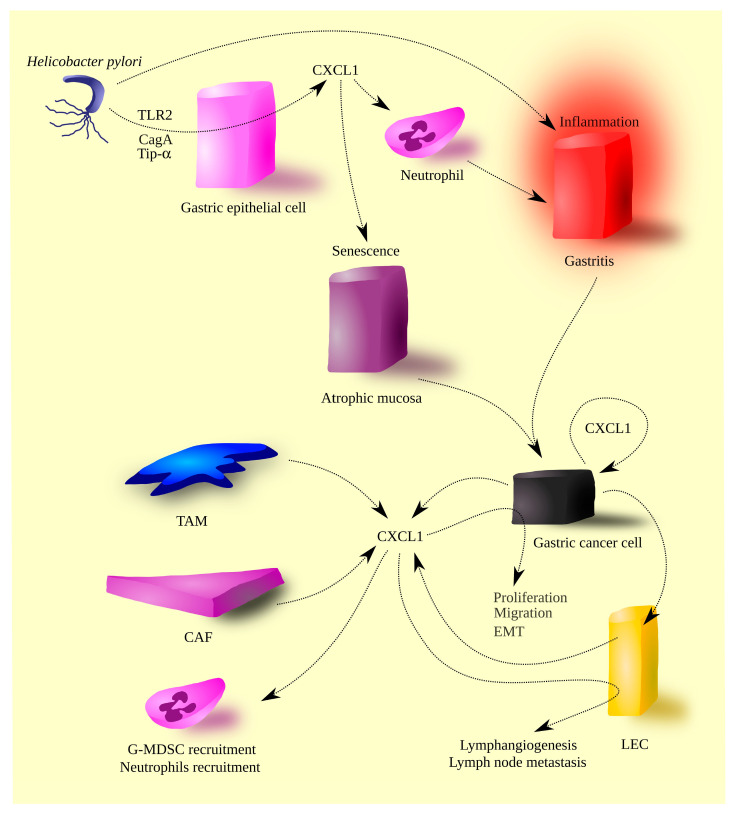
The role of CXCL1 in gastric cancer pathogenesis. *Helicobacter pylori* infection induces inflammatory reactions within the gastric wall, leading to gastritis and atrophic mucosa. These sequelae of *H. pylori* infection are partially mediated by CXCL1. The bacterium upregulates this chemokine’s expression through its virulence factors (Tip-α and CagA) and activation of TLR2. The persistent inflammatory state culminates in gastric cancer. Gastric cancer cells express CXCL1, which in turn amplifies its own production. Other sources of CXCL1 in gastric tumors include tumor-associated macrophages (TAMs), cancer-associated fibroblasts (CAFs), and lymphatic endothelial cells (LECs). CXCL1 participates in tumorigenic processes, promoting the migration of gastric cancer cells. Furthermore, CXCL1 facilitates the recruitment of granulocytic myeloid-derived suppressor cells (G-MDSCs) and neutrophils. CXCL1 also influences LECs, resulting in lymphangiogenesis and the migration of gastric cancer cells to lymphatic vessels, ultimately leading to lymph node metastasis.

**Table 1 cells-12-01406-t001:** The effect of CXCL1 expression level on the survival of patients with head and neck cancer.

Type of Tumor	Effect on the Survival at High CXCL1 Expression	Number of Patients in the Study	Notes	References
Head and neck cancer: squamous cell carcinoma	Worse prognosis	491	OS, DFS Data from the Cancer Genome Atlas (TCGA) dataset	[31,52]
Head and neck cancer: squamous cell carcinoma	Worse prognosis	494	OS, DFS/PFS Data from TCGA dataset	[51,52]
Head and neck cancer: squamous cell carcinoma	Worse prognosis	499	OS, statistically insignificant difference in RFS Data from Kaplan–Meier plotter	[26,27]
Head and neck cancer: larynx squamous cell carcinoma	Worse prognosis	135	OS	[33]

DFS—disease-free survival; OS—overall survival; PFS—progression-free survival; RFS—relapse-free survival; red color—worse prognosis.

**Table 2 cells-12-01406-t002:** The effect of CXCL1 expression level on the survival of patients with esophageal cancer.

Type of Tumor	Effect on the Survival at High CXCL1 Expression	Number of Patients in the Study	Notes	References
Esophageal cancer: esophageal squamous cell carcinoma	Worse prognosis	141	OS,	[56]
Esophageal cancer: esophageal squamous cell carcinoma	Better prognosis	877	OS Data from the Kaplan–Meier Plotter database (http://kmplot.com access date: 16 April 2022)	[57]
Esophageal carcinoma	No significant impact on prognosis	92	RFS and OS Data from the gene expression profiling interactive analysis (GEPIA) database	[65]

OS—overall survival; RFS—relapse-free survival; red color—worse prognosis; blue color—better prognosis.

**Table 3 cells-12-01406-t003:** The effect of CXCL1 expression level on the survival of patients with gastric cancer.

Type of Tumor	Effect on the Survival at High CXCL1 Expression	Number of Patients in the Study	Notes	References
Gastric cancer	Worse prognosis	98	OS	[98]
Gastric cancer	Worse prognosis	56	Cumulative survival, CXCL1 in the tumor, as well as plasma CXCL1 level	[97]
Gastric cancer	Better prognosis	34	OS	[96]
Gastric cancer	Worse prognosis	572	OS, Data from the public database	[108]
Gastric cancer	Worse prognosis	155	OS	[107]
Gastric cancer	Worse prognosis	127	OS	[115]
Gastric cancer	Worse prognosis	590	OS	[101]
Gastric cancer	Worse prognosis	100	OS	[105]
Gastric cancer	Worse prognosis	263	OS, Only for stage I patients	[112]
Gastric cancer	Worse prognosis	105	OS	[109]
Gastric cancer	Worse prognosis	72	OS	[99]

OS—overall survival; red color—worse prognosis; blue color—better prognosis.

**Table 4 cells-12-01406-t004:** The effect of CXCL1 expression level on the survival of patients with liver cancer.

Type of Tumor	Effect on the Survival at High CXCL1 Expression	Number of Patients in the Study	Notes	References
Liver cancer	Worse prognosis	346	OS, Data from TCGA dataset	[52,151]
Liver cancer: HCC	Worse prognosis	182	OS Data from the Kaplan–Meier plotter database	[152]
Liver cancer: HCC	Worse prognosis	48	OS	[131]
Liver cancer: HCC	Worse prognosis	119	OS, DFS	[143]
Liver cancer: HCC	Worse prognosis	259	OS, RFS Patients with a high CXCL1 expression in a tumor together with a high CXCR2 expression	[147]

DFS—disease-free survival; OS—overall survival; RFS—relapse-free survival; red color—worse prognosis.

**Table 5 cells-12-01406-t005:** The effect of CXCL1 expression level on the survival of patients with cholangiocarcinoma.

Type of Tumor	Effect on the Survival at High CXCL1 Expression	Number of Patients in the Study	Notes	References
Cholangiocarcinoma	Better prognosis	165	OS	[155]
Cholangiocarcinoma	Worse prognosis	18	RFS No effect at OS, Data from the GEPIA database	[65]

OS—overall survival; RFS—relapse-free survival; red color—worse prognosis; blue color—better prognosis.

**Table 6 cells-12-01406-t006:** The effect of CXCL1 expression level on the survival of patients with pancreatic cancer.

Type of Tumor	Effect on the Survival at High CXCL1 Expression	Number of Patients in the Study	Notes	References
Pancreatic cancer	Worse prognosis	160	OS	[163]
Pancreatic cancer: pancreatic adenocarcinoma	No significant impact on prognosis	178	OS, DFS	[165]
Pancreatic adenocarcinoma	No significant impact on prognosis	90	OS, RFS Data from the GEPIA database	[65]

DFS—disease-free survival; OS—overall survival; RFS—relapse-free survival; red color—worse prognosis.

**Table 7 cells-12-01406-t007:** The effect of CXCL1 expression level on the survival of patients with colorectal cancer.

Type of Tumor	Effect on the Survival at High CXCL1 Expression	Number of Patients in the Study	Notes	References
Colorectal cancer	Worse prognosis	62	OS	[238]
Colorectal cancer	Worse prognosis	91	RFS, analysis only in stage III patients. In other stages, CXCL1 expression is not related to prognosis	[204]
Colorectal cancer	No significant impact on prognosis	163	RFS, OS analysis only in stage II patients	[204]
Colorectal cancer	No significant impact on prognosis	270	OS, Data from the GEPIA database	[65,206]
Colorectal cancer	No significant impact on prognosis	362	RFS, OS Data from the GEPIA database	[65,239]
Colorectal cancer	Worse prognosis	276	OS, DFS	[233]
Colorectal cancer	No significant impact on prognosis	125	OS, RFS	[212]
Colorectal cancer	Worse prognosis	45	OS, only stage IV patients	[218]
Colorectal cancer	No significant impact on prognosis	70	OS, Only stage II and III patients	[218]
Colorectal cancer: colon cancer	Better prognosis	438	OS, data from TCGA	[52,209,213]
Colorectal cancer: colon adenocarcinoma	No significant impact on prognosis	171	OS, RFS	[209]
Colorectal cancer: rectal adenocarcinoma	Worse prognosis	304	OS Data from the TCGA database	[52,229]

DFS—disease-free survival; OS—overall survival; RFS—relapse-free survival; red color—worse prognosis; blue color—better prognosis.

**Table 8 cells-12-01406-t008:** CXCR2 inhibitors in selected clinical trials. Source: ClinicalTrials.gov NIH U.S. National Library of Medicine website.

Disease for Which the Drug Is Being Tested	Drug Name	Clinical Trial Phase	ClinicalTrials.gov Identifier
Metastatic Castration-Resistant Prostate Cancer	AZD5069	I and II	NCT03177187
Myelodysplastic Syndromes	SX-682	I	NCT04245397
Pancreatic Ductal Adenocarcinoma	SX-682	I	NCT04477343
Melanoma Stage III and Stage IV	SX-682	I	NCT03161431
Inflammatory Response	RIST4721	I	NCT04105959
Chronic Obstructive Pulmonary Disease (COPD)	Danirixin (GSK1325756)	I	NCT01453478
COPD	Danirixin (GSK1325756)	I	NCT03136380
COPD	Danirixin (GSK1325756)	II	NCT03250689
COPD	Danirixin (GSK1325756)	II	NCT02130193
COPD	Navarixin (SCH 527123, MK-7123)	II	NCT01006616
COPD	QBM076	II	NCT01972776
Influenza	Danirixin (GSK1325756)	II	NCT02469298
Influenza	Danirixin (GSK1325756)	II	NCT02927431
Respiratory Syncytial Virus (RSV) Infections	Danirixin (GSK1325756)	I	NCT02201303
COPD	SB-656933	I	NCT00504439
Cystic Fibrosis	SB-656933	II	NCT00903201
Type 1 Diabetes	Ladarixin	II	NCT05035368
Type 1 Diabetes	Ladarixin	III	NCT04628481
Bullous Pemphigoid	DF2156A	II	NCT01571895

## Data Availability

Not applicable.

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
