# Peer review of "The Clinical Significance and Role of CXCL1 Chemokine in Gastrointestinal Cancers"

_cells, 2023, doi:10.3390/cells12101406_

Round 1

Reviewer 1 Report

This review paper describes the various roles of CXCL1 chemokine in various gastrointestinal cancers. The review paper covers much research related to CXCL1. As indicated in the paper, somebody has to elucidate the accomplishment of CXCL1 research. Therefore, this paper is a meaningful review paper. This review paper also presents mixed results and the complicated roles of CXCL1 in cancer biology. Therefore, the review also recommends adding the future perspective related to CXCL1 research to the end of the manuscript. The reviewer also asks the authors to correct the following errors.

1. When the authors express numbers, they used 600,000 and 544 thousand together throughout the manuscript. (For example: Lines 130 and 131). Please unify to one expression.

2. Line 45: intercellular signaling within tumor cells --> intercellular signaling within tumor tissue? "Intercellular signal" cannot be "signaling within tumor cells".

3. Table 1-7: Please provide the following terms. TGCA, GEPIA.

4. The authors used "/" to express many functions, confusing readers. For example, another name for CXCL8 is IL-8. Therefore, they do not have to express CXCL8 as CXCL8/IL-8. In some sentences, they also use"/" to express CXCL1/CXCL8-CXCR2 in Line 247. In this case, CXCL1 is not CXCL8. Therefore, the reviewer suggests expressing the following way. For example, Line 215: CCL3/macrophage inflammation protein 1a (MIP-1a) --> macrophage inflammation protein 1a (MIP-1a, CCL3)

5. Line 210: outer inflammatory protein --> outer inflammatory A protein

6. Line 211: vacuolating cytotoxin (Vac A) --> vacuolating cytotoxin A (Vac A) 

7. Line 229: The full name of TLR2 has to be defined in Line 226.

8. Line 231: TNF-a is defined in Line 160.

9. Line 236: Tip-a is defined in Line 233.

10. Line 233-235: Please change various chemokine definitions as follow. CXCL10/g interferon inducible protein 10 (IP-10) --> g interferon inducible protein 10 (IP-10, CXCL10)

11. Line 261: PAR1b (Full name?)

12. Line 267: MDM2 (Full name?)

13. Line 281: miR-204 --> microRNA (miR)-204

14. Line 283: The full name of TAM has to be defined in Line 153.

15. Line 301: LEC  is defined in Line 101.

16. Line 304: Please provide a reference.

17. In the liver cancer section: please change all "hepatocellular carcinoma"   to HCC because you already define HCC in Line 32.

18. Line 360: S100A8/9 (Full name?)

19. Line 377: aSMA (Full name?)

20. Line 422: mTORC1 (Full name?)

21. Line 438: CCL17/TARC --> thymus- and activation-regulated chemokine (TARC, CCL17)

22. Line 465: MSC (Full name?)

23. Line 512: ApoE is defined in Line 510

24. Line 561: CXCL5/ENA78 --> epithelial-derived neutrophil-activating peptide 78 (ENA78, CXCL5)

25. Line 567-568: atypical chemokine receptor 1 (ACKR1)/Duffy antigen receptor for chemokines (DARC) --> atypical chemokine receptor 1 (ACKR1, Duffy antigen receptor for chemokines, DARC)

26. Line 588: TAN --> TAM?

27. Line 645: HIF-2 (Full name?)

28. Line 648: cell migration inducing hyaluronidase 1 (CEMIP)/KIAA1199 --> cell migration inducing hyaluronidase 1 (CEMIP, KIAA1199)

29. Line 649: TGFBR1/2 (Full name?)

30. Line 660: SDF-1 has to be defined in Line 291

31. Line 733: RBE (Full name?) 

Author Response

Rev.1.

The manuscript "The Clinical Significance and Role in Molecular Processes of CXCL1 Chemokine in Gastrointestinal Cancers: Head and Neck Cancer, Esophageal Cancer, Gastric Cancer, Liver Cancer, Cholangiocarcinoma, Pancreatic Cancer, and Colorectal Cancer" submitted makes a significant contribution to research by addressing the gap with a detailed analysis of CXCL1's role in gastrointestinal cancers. This review also summarizes the CXCL1's potential as a therapeutic target in anti-cancer therapy.

Comments/Suggestions to Authors:
1. I would suggest changing the title to: “Clinical Significance and Role of CXCL1 Chemokine in Gastrointestinal Cancers”. Keep the title short, concise, and broad. No need to specify all these different forms of gastrointestinal cancers in the title. Possibly, I would advise (if the keyword limit permits) including those various cancers in the "keywords" section to improve visibility.

  1. Abstract is well-written, simple to understand for every reader.
    3. In Introduction session, please provide reference for this sentence “The CXC chemokine subfamily includes 16 representatives in humans ........... followed by CXCL1”.
    4. It's essential to bridge/address the gap, and the authors did a good job at it in the introduction session.
    5. There seems to be inadequate Background/Introduction section. It's nice to learn about gastrointestinal cancer cases, however I believe there should be more information included about the CXC chemokine family as well, with a focus on CXCL1. Please review the following references and add some more information to the introduction section:

https://www.ncbi.nlm.nih.gov/pmc/articles/PMC7501593/
https://www.mdpi.com/2072-6694/15/1/167
6. If authors could include a table listing the CXCR2 antagonists studied in clinical trials that would be good.
7. Readers will be interested if a graphical summary (like an overview) is included to explain CXCL1's involvement or role in gastric cancer.
8. The conclusion of this review should be put at the end in a single paragraph, along with future perspectives

The article has been revised according to the reviewer's recommendation.

Reviewer 2 Report

Please find the attached PDF. 

Author Response

Rev.2.

This review paper describes the various roles of CXCL1 chemokine in various gastrointestinal cancers. The review paper covers much research related to CXCL1. As indicated in the paper, somebody has to elucidate the accomplishment of CXCL1 research. Therefore, this paper is a meaningful review paper. This review paper also presents mixed results and the complicated roles of CXCL1 in cancer biology. Therefore, the review also recommends adding the future perspective related to CXCL1 research to the end of the manuscript. The reviewer also asks the authors to correct the following errors.

  1. When the authors express numbers, they used 600,000 and 544 thousand together throughout the manuscript. (For example: Lines 130 and 131). Please unify to one expression.
  2. Line 45: intercellular signaling within tumor cells --> intercellular signaling within tumor tissue? "Intercellular signal" cannot be "signaling within tumor cells".
  3. Table 1-7: Please provide the following terms. TGCA, GEPIA.
  4. The authors used "/" to express many functions, confusing readers. For example, another name for CXCL8 is IL-8. Therefore, they do not have to express CXCL8 as CXCL8/IL-8. In some sentences, they also use"/" to express CXCL1/CXCL8-CXCR2 in Line 247. In this case, CXCL1 is not CXCL8. Therefore, the reviewer suggests expressing the following way. For example, Line 215: CCL3/macrophage inflammation protein 1a (MIP-1a) --> macrophage inflammation protein 1a (MIP-1a, CCL3)
  5. Line 210: outer inflammatory protein --> outer inflammatory A protein
  6. Line 211: vacuolating cytotoxin (Vac A) --> vacuolating cytotoxin A (Vac A) 
  7. Line 229: The full name of TLR2 has to be defined in Line 226.
  8. Line 231: TNF-a is defined in Line 160.
  9. Line 236: Tip-a is defined in Line 233.
  10. Line 233-235: Please change various chemokine definitions as follow. CXCL10/g interferon inducible protein 10 (IP-10) --> g interferon inducible protein 10 (IP-10, CXCL10)
  11. Line 261: PAR1b (Full name?)
  12. Line 267: MDM2 (Full name?)
  13. Line 281: miR-204 --> microRNA (miR)-204
  14. Line 283: The full name of TAM has to be defined in Line 153.
  15. Line 301: LEC  is defined in Line 101.
  16. Line 304: Please provide a reference.
  17. In the liver cancer section: please change all "hepatocellular carcinoma"   to HCC because you already define HCC in Line 32.
  18. Line 360: S100A8/9 (Full name?)
  19. Line 377: aSMA (Full name?)
  20. Line 422: mTORC1 (Full name?)
  21. Line 438: CCL17/TARC --> thymus- and activation-regulated chemokine (TARC, CCL17)
  22. Line 465: MSC (Full name?)
  23. Line 512: ApoE is defined in Line 510
  24. Line 561: CXCL5/ENA78 --> epithelial-derived neutrophil-activating peptide 78 (ENA78, CXCL5)
  25. Line 567-568: atypical chemokine receptor 1 (ACKR1)/Duffy antigen receptor for chemokines (DARC) --> atypical chemokine receptor 1 (ACKR1, Duffy antigen receptor for chemokines, DARC)
  26. Line 588: TAN --> TAM?
  27. Line 645: HIF-2 (Full name?)
  28. Line 648: cell migration inducing hyaluronidase 1 (CEMIP)/KIAA1199 --> cell migration inducing hyaluronidase 1 (CEMIP, KIAA1199)
  29. Line 649: TGFBR1/2 (Full name?)
  30. Line 660: SDF-1 has to be defined in Line 291
  31. Line 733: RBE (Full name?) 

Note 4 - According to the current classification, chemokines are named based on the CC or CXC motif at the N-terminus and a number. Therefore, we have used such chemokine names in our work

Zlotnik A, Yoshie O. Chemokines: a new classification system and their role in immunity. Immunity. 2000 Feb;12(2):121-7. doi: 10.1016/s1074-7613(00)80165-x

Note 26 – The term TAN has been introduces.

Note 31 - This is the name of the cell line, and even the scientists who derived this line did not disclose why they named these cells as such.

Enjoji M, Sakai H, Nawata H, Kajiyama K, Tsuneyoshi M. Sarcomatous and adenocarcinoma cell lines from the same nodule of cholangiocarcinoma. In Vitro Cell Dev Biol Anim. 1997 Oct;33(9):681-3. doi: 10.1007/s11626-997-0125-z
